# Proteomics Methodologies: The Search of Protein Biomarkers Using Microfluidic Systems Coupled to Mass Spectrometry

**DOI:** 10.3390/proteomes11020019

**Published:** 2023-05-10

**Authors:** Isabel De Figueiredo, Bernard Bartenlian, Guillaume Van der Rest, Antoine Pallandre, Frédéric Halgand

**Affiliations:** 1Institut de Chimie Physique, Université Paris Saclay, Avenue Jean Perrin, F91400 Orsay, France; isabel-ines.de-figueiredo-goncalves@universite-paris-saclay.fr (I.D.F.); antoine.pallandre@universite-paris-saclay.fr (A.P.); 2Centre des Nanosciences et Nanotechnologies, Université Paris Saclay, 10 Boulevard Thomas Gobert, F91120 Palaiseau, France; bernard.bartenlian@universite-paris-saclay.fr

**Keywords:** pre-treatment sample, biomarkers, chromatography, microfluidics, mass spectrometry (MS)

## Abstract

Protein biomarkers have been the subject of intensive studies as a target for disease diagnostics and monitoring. Indeed, biomarkers have been extensively used for personalized medicine. In biological samples, these biomarkers are most often present in low concentrations masked by a biologically complex proteome (e.g., blood) making their detection difficult. This complexity is further increased by the needs to detect proteoforms and proteome complexity such as the dynamic range of compound concentrations. The development of techniques that simultaneously pre-concentrate and identify low-abundance biomarkers in these proteomes constitutes an avant-garde approach to the early detection of pathologies. Chromatographic-based methods are widely used for protein separation, but these methods are not adapted for biomarker discovery, as they require complex sample handling due to the low biomarker concentration. Therefore, microfluidics devices have emerged as a technology to overcome these shortcomings. In terms of detection, mass spectrometry (MS) is the standard analytical tool given its high sensitivity and specificity. However, for MS, the biomarker must be introduced as pure as possible in order to avoid chemical noise and improve sensitivity. As a result, microfluidics coupled with MS has become increasingly popular in the field of biomarker discovery. This review will show the different approaches to protein enrichment using miniaturized devices and the importance of their coupling with MS.

## 1. Introduction

Ground-breaking advances in the field of biotechnologies have allowed for the “birth” of many analytical tools without which progress in many fields such as the medical, pharmaceutical [1,2], environmental, and foodstuff progress [3,4] fields would be scarcely imaginable [5]. Most importantly, these advances play a significant role both in the quantification and identification of molecular biomarkers in diseases, facilitating the development of more efficient diagnosis methods, leading to better-targeted medications, and therefore to better treatments [6,7]. These advances require highly sophisticated and efficient analytical workflows, including sample preparation, target capture, elution, separation, and quantitative/qualitative detection [8,9]. One of the most important aspects and the first to be accomplished to characterize a target protein (required for many experimental applications, including structural studies and in vitro biochemical assays), is its separation from all other proteins or molecules present in a proteome. To this end, various chromatographic methods have been developed [10]. Indeed, many publications reported the description of ion-exchange [2], size-exclusion [8,11], affinity [12], and adsorption-based chromatography [13], as well as their performance and optimal design. Several reviews have also described the fundamental mechanisms, operations, and applications of these chromatographic techniques to that end [8,14]. However, the analysis of proteins through these methods is often accompanied by the exposition of the target to several stresses such as strong acids used as ion-pairing agents or organic solvents used for elution of proteins, contact with different materials (e.g., adsorption to the resin) [15], temperature [16], shear stress [17], and ionic strength [18]. These stresses can significantly alter the protein structure [19], leading to a loss of its native functionality and interaction with other molecules [8]. From a diagnostic and therapeutic point of view, the structural alteration of the native protein during the analytical procedure can threaten the design of safe and efficient diagnosis. In this way, a compromise between separation performance and preservation of the protein native structure must be achieved.

Size-exclusion chromatography (SEC) and affinity chromatography (AC) have been historically used and offer different benefits, especially for the purification of therapeutic molecules due to the possibility of using specific conditions (physiological buffer, pH, ionic strength, etc.) that result in diminished effects on the structure of molecules and the local environment [8,15,20]. Among the different chromatographic techniques used today, SEC is the choice for the size-based fractionation of biomacromolecules, due to the good and non-destructive separation of large molecules from small molecules [8,11,15]. Likewise, AC is a technique based on biological functions that allows for the purification of proteins from complex media. This is carried out through a highly specific interaction between the ligand and the protein of interest, conferring a high selectivity [21]. Unfortunately, several points associated with experimental conditions need to be optimized, such as elution agents, matrix support characteristics, and biological affinity pair (antigen–antibody) [20,21]. Beyond that, in some cases we have limited quantities of the target biomolecule available that are massively diluted with unspecific molecules present in higher concentrations, thus requiring high-efficiency purification or separation processes of the target biomolecules. Generally, biomarkers are present in low abundance, especially at the outset of disease. If discovery and quantification could be easily performed, this would potentially promote effective diagnosis, usually far earlier in the progression of illness. Consequently, the setup of these methods has changed. Today, an attractive option is miniaturization, called by many “the new era of the analytical sciences”. Miniaturization has led, in the past two decades, to revolutionary advances in terms of high-throughput analysis and point-of-care (POC) applications for their reduced cost, as compared to conventional macroscale laboratory techniques [7,22,23]. 

In this state of the art, two standard biochemical techniques, SEC and AC, used for protein fractionation along with an overview of these innovative approaches are discussed. In addition, some advances that have been verified in this area and how they can improve the diagnosis and treatment of certain diseases in the future such as Parkinson’s disease (PD) are also summarized. Here, we also overview the progress that has been made in the coupling of lab on chip electrospray ionization mass spectrometry (chip-ESI-MS) in order to improve the small-volume bio-specimen analysis (such as blood and urine), important for biomarker discovery and thus, clinical settings. 

## 2. Methodologies for Fractionation of Proteins under Native Condition

A biological sample is quite complex and non-Newtonian in nature, exhibiting many different physicochemical dimensions such as a large range of molar concentrations, sizes, the nature of compounds, and molecular structural dynamics. In the case of the blood, for example, although it is easy to sample and very reliable to diagnose certain diseases and physiological states, the existence of a huge array of different proteins in its compositions, each one with a different concentration range and degree of interaction, enhances the complexity of detection in this proteome, especially for proteins that exist at low concentrations. Several approaches have been employed for protein separation including precipitation, centrifugation, and electrophoresis to reduce this complexity, including the elimination of haemoglobin in blood [24]. In fact, protein separation is a decisive first step to decrease this complexity. However, the separation of these compounds is not an easy task in requiring all possible precautions to be taken. Certain analytes, including proteins, are easily degraded, losing key chemical and physical characteristics, and eventually leading to a conformational change [25,26]. Accordingly, in regard to the study of native conformation and biological activity of compounds, the recourse to some of these separation methods including precipitation and centrifugation is not ideal. Analytes are subjected to aggressive environments, resulting in considerable structural change, and are consequently a problem for structural diagnostics.

Since the beginning of the 2000s, common physical and chemical techniques, including X-ray diffraction (XRD), nuclear magnetic resonance (NMR), and Förster resonance energy transfer (FRET), have provided a deep comprehension of protein properties, especially in the detection of protein conformational changes in vitro and in situ. XRD can supply a high resolution of static structures, but the activity of the crystalline macromolecule can only be implicit. On the other hand, NMR provides powerful insights into the relationship between structure and biological function, but its application is mainly limited to proteins that do not exceed 30 kDa and it demands considerable in protein amounts [27]. FRET is used to follow protein conformational change in real time, but the instability in fluorescence emission and photobleaching of the fluorophores may limit the spatial accuracy and temporal resolution. In addition, the use of fluorescent labels that can deform the target analyte structure and the limitation of determining an atomic-level three-dimensional structure can become a weakness for structural studies [27,28]. Electrophoresis is also commonly used for protein characterization and separation that is essentially based on its size, but also its charge and shape (depending on the type of electrophoresis used). Mainly two types of electrophoresis exist: sodium dodecyl sulphate polyacrylamide gel electrophoresis (SDS-PAGE) and native gel electrophoresis. The two differ essentially in that in first one, the separation is based upon the molecular size in denaturing conditions and in the second one, the separation is based upon on the size, shape, and charge of the protein. Even though both techniques are widely applied in a variety of proteomic studies, electrophoresis most often fails in the analysis of low-abundance proteins [29]. Immuno-based systems such as enzyme-linked immunosorbent assay (ELISA) and Western blot (WB) have also been studied and employed in a large scale for protein characterization. However, these methods require high-quality antibodies [30], which are difficult to obtain due to potential lot-to-lot variability [31] and cross-reactivity of the antibodies [30,32,33,34] that often weaken both assay performance in terms of limit of detection (LOD) and experimental bias. In addition, both methods comprise numerous and demanding steps, such as long incubation times, the use of considerable volumes of buffers, and costly reagents [22,34,35]. Finally, liquid chromatography (LC) is a powerful method for working with biological samples and one of the most effective and broadly used one for protein separation/purification purposes. Due to the progression conducted in LC, there is a strong trend in recent years to incorporate these techniques for protein separation without affecting their conformational structure, and consequently their biological activity [21].

### 2.1. Chromatography-Based Techniques for Protein Separation: SEC and IAC

Historically, the first chromatographic experiences were developed in 1901 by the botanist Tsvet during his research on plant pigments [36]. Using a narrow glass tube packed with powdered chalk, he observed different bands from a plant extract [37]. The technique has rapidly advanced and been refined over the years, where the columns were filled with superficially porous particles (~50 µm in diameter) into 1 mm glass columns, becoming identified as high-performance liquid chromatography (HPLC) [38]. Although these columns have shown a gain in efficiency, the low surface area given by the thin layer of porous particles was limited compared with the larger porous particles. As a consequence, a larger concentration of samples was needed for low-sensitivity detectors [38]. Further work focused on controlling porosity, particle size, and rigidity that influenced the efficiency of the separation. Particularly, small and well-packed particles in a column lead to higher efficiencies and consequently, higher resolutions. However, the increased pressure necessary for HPLC can be a major obstacle if users’ samples are not amenable to being used at high pressures. The introduction of rigid porous polymer monolithic materials in the early 1990s [39] has been studied and optimized to the extent that nowadays monolithic columns characterized by a single piece of highly porous material such as chromatography support enables higher linear flow rates over the packed ones, leading to the faster mass transfer of molecules, especially relevant for large molecule separation where diffusion is slow [40]. In addition, the use of rigid monolithic columns lessens the use of high pressure that occurs when the particle size is reduced, an important factor since the decrease in particle sizes provides better column efficiency [40,41]. Moreover, their great flexibility in terms of chemical functionalization [42] and morphology [43] that can be easily adjusted makes them a suitable tool for predestined proteomic applications.

Soft gels, such as dextran-based SephaDex and polyacrylamide-based BioGel P6, that are characterized by a higher internal porosity compared to more rigid packing particles such as silica, have contributed to a higher selectivity and analysis of larger sample volumes. However, their low mechanical strength limits their employment in high-pressure conditions. Even so, BioGel P6 proved to be the most appropriate resin for flow rates of tens of µL/min [44]. These types of porous chromatographic stationary phases have been subjected to intensive use. SEC, also entitled as advanced particle characterization or gel permeation, is a size-based technique: the molecules do not bind to the resin since this technique does not depend on interactions with the stationary phases. Instead, the molecules of either proteins or polymers are separated according to their physical size (hydrodynamic volume), with the large molecules eluting more rapidly through the column compared to the small molecules that are instead trapped into the porous beads, and thus take more time to elute. This technique has been deemed to be a robust method for a wide type of protein fractionations due to its speed, reproducibility, and ease in the separation and structural characterization of protein and aggregates [8,45], a benefit when it comes to the study of structural diseases [46]. It can be appropriate for reducing sample complexity, quick desalting, and buffer exchange. 

The fusion between LC and antibody (Ab)-specific binding, named as immuno-affinity chromatography (IAC), has also become attractive, competitive, and commonly used [47], especially as a final step for a particular target purification. For IAC, different matrixes are used in order to embed antibodies within columns. Among many, natural polymer (e.g., agarose and cellulose) and synthetic organic supports (e.g., acrylamide polymers, copolymers, or derivatives) stand out. Despite being attractive for their low cost, they usually work at low pressure [47]. 

### 2.2. Immunopurification

The use of column-free magnetic nanoparticles (NPs) has increasingly been recognized as a recourse method exhibiting high binding capacity [20]. This approach, called by many as immunopurification (IP), is based on the use of functionalized NPs with protein binders such as protein A/G [47] or aptamer/oligonucleotides [48] that represent an emerging class of ligands with a high ability in analyte capture and release of antigen/antibody complexes. Under magnetic forces, magnetic beads pre-coated with these binders that form a complex with Ab (generally Immunoglobulin G) engage the use of their specific interactions to capture a target analyte (Ag) while the suspending liquid is removed [20]. The quality of the assay is hard to predict as it heavily depends on affinity between the ligand and the target analyte, the concentration, the incubation time, the temperature, and elution strength that will affect or improve the efficiency of the detection of the Ag in the solution [20,49]. These issues withstanding, a good number of studies have nevertheless been reported using IP to essentially purify proteins or protein complexes. For example, Biasini and colleagues [50] managed to purify aggregated full-length prion protein isoforms (PrPSc) from different mouse models of prion diseases by centrifugation and immunoprecipitation. The monoclonal antibody which was used was able to distinguish the structural features common to infectious forms of misfolded prion proteins versus non-infectious ones. Following the same immunocapture principle to catch the progastrin-releasing peptide, a low-abundance biomarker for small cell lung cancer, Levernæs et al. used magnetic beads functionalized with anti-progastrin antibodies [51]. After successful peptide capture, the eluates were directly injected into a nano LC-MS system for the quantification of progastrin-releasing peptide (ProGRP). The peptide extraction presented more pure extracts and less matrix effects than the extraction of intact proteins. 

Together with the sample treatment, MS has been the analytical read-out method of choice capable of identifying and differentiating proteins and peptides. Despite being commonly used, MS essentially suffers from two major specific challenges, that refer to the incompatibility of some elution media with MS [47,52] and the poor sensitivity when analyzing some low-abundance biomarkers in complex samples [53]. Hence, there is a need to prepare and enrich the sample as best as possible without losing or structurally disturbing the biomarker of interest [54]. Chemicals such as buffers, salts, reducing agents, and detergents used in SEC, AC, and IP are generally crucial for sample preparation and analysis. These reagents (e.g., NaCl) frequently used in SEC interfere with the detection of the protein by MS, since salts are not compatible with this type of detector. Thus, there is a need for replacing them before proceeding to MS analysis [55] with other compatible mobile phases such as volatile salts (e.g., ammonium acetate (AA) and triethylammonium acetate (TEAA) [47]. Interestingly, mobile phases used to improve SEC protein separation have risen a point of debate in the scientific community due to their potential effect on protein conformation [15,56,57]. For this reason, protein separation/purification conditions must be carefully determined to achieve a compromise between separation resolution, signal intensity, and loss of proteins structure or formation of aggregates [10]. Another point which is usually forgotten, is that the detection of protein isoforms can potentially be used as disease biomarkers [58]. Protein isoforms found, for example, in neurodegenerative disorders including Tau protein [59] present a high percentage of amino acid sequence homology. Thus, regarding the characterization of these isoforms (e.g., protein mutants), a technique must be highly specific. Two complementary points need to be explored towards this aim: (1) enhancing chromatographic separation (in terms of resolution, plate number, reproducibility, repeatability) and (2) choosing the appropriate detection method and parameters in order to characterize and quantify protein isoforms [21,58]. Recent developments in the microfluidic field in combination with ion mobility (IM) and MS have been proved as promising POC approaches for the detection of these protein isoforms. The ion mobility principle and its applications are detailed below.

## 3. Microfluidics

Microfluidics has gained substantial attention in the last years and are considered a potential future strategy in the discovery of new biomarkers and drugs for clinical uses revolutionizing medicine and pharmacy. Microfluidics allows the reduction of sample volume consumption, thereby limiting the amount of biological sample volume used (blood, cerebrospinal fluid (CSF)) [23,35,60] in agreement with clinical requirements. These biomimetic microsystems are leading to the emergence of new “low-cost” techniques that break from the traditional biochemical techniques. Likewise, the miniaturization of different biochemical techniques makes it possible to conceive an environment that is more comparable to physiological states (e.g., mimicking blood vessels) [61]. At the same time, they have attracted significant attention by their fast analysis time, potential automation, high-throughput capabilities, possibility of integrating different functional modules in one system, portability, and high resolution of analysis as compared to the conventional instrumentation currently widely established in academic and industrial settings [23]. An additional point that microfluidics offers over macroscale devices is a laminar flow regime due to low Reynolds (Re) numbers (where viscous forces are dominant over inertial forces). Microfluidics allows for a better control over the flow variables, i.e., velocity, shear stress, temperature, and chemical concentrations. This controlled environment is especially important for protein structure preservation. Under laminar flow conditions, microfluidics can provide a 3D biomimetic environment, where mechanical forces including shear stress need to be scrutinized. Fluid shear stress is determined by fluid velocity and viscosity. In capillary systems, wall shear stress can have detrimental effects when examining sensitive biological analytes, especially in terms of structure. Recently, Hakala et al. demonstrated that the shear stress caused by capillary transport induced local protein structural changes, and thus flow control is a crucial constant in order to control shear stress [62]. The microdevices are designed with several inlet ports where the sample is usually introduced via tubes, and also with several reservoirs to pre-store the required buffers [63]. Due to laminar flow behaviour (low Re), microfluidics often suffers from limited mixing [63], for instance, limiting the analyte capture in immunocapture as there is not a sufficient contact time between the analyte and the Ab for effective interaction. Distinct strategies were found to overcome this problem. The design of a serpentine or spiral-shaped channel in microfluidics has been advantageously used to increase the mixing of the injected solutions [64]. Another strategy to increase the mixing of the solutions was presented by Pereiro and his colleagues [65]. They displayed a fluidized bed configuration, where they replaced gravitational forces by magnetic forces. Subsequently, beyond a certain threshold pressure, the fluidization of magnetic beads (in the size range from 1 to 5 μm) occurred, and an expansion of the bed was observed. Plus, they demonstrated that the magnetic streamline and field gradient of the magnet is the major parameter in the fluidization phenomena, where higher angles lead to smaller clusters and thus higher porosity, whereas on the contrary, small angles resulted in no fluidization (see Figure 1).

This can be particularly interesting in the immunocapture of proteins as it provides a better interaction between the Ab functionalized magnetic beads with the target molecule. Subsequently, they tested this microfluidics system for biomarker capture and pre-concentration, obtaining favourable results: an LOD of 0.2 ng min^−1^ for 200 μL of sample volume and a capture efficiency around 85% [65]. Since the flow rate (1 to 4 µL/min) used in microfluidics is close to the ionization source flow rates, the chip can potentially be directly coupled to MS (detailed later, Section 5). 

Separation of analytes by size was performed by the introduction of membranes/gel-based materials inside the channel [63,66]. The integration of membranes provides a number of advantages, especially with regards to the recovery of biomacromolecules. However, the formation of clotting in these membrane systems can be problematic [44,63]. In order to address this problem, recently Chen and his colleagues [67] employed a membrane-based filtration with a stirring strategy (a vacuum and then a vortex flow generated by a micro-stirrer) continuously agitating the blood cells and platelets to alleviate clogging problems. The introduction of cross-linked gels, such as polyacrylamide (PA), has also been applied for on-chip chromatography and protein sizing, since this kind of sieve matrix intensifies mobility differences among molecules and can be improved by tuning the pore size. Hou and Herr [68] presented an interesting work where they introduced a discontinuous PA gel (with an increasing percent of acrylamide along the gel) in order to enhance Ab and complex mobility differences when compared to a uniform gel, hence improving the electrophoretic separation of proteins. In addition, this ultra-short separation allowed for a reduced application of electrical potentials. Their results were greatly satisfying with a lower limit of detection of 11 ng/mL for C reactive protein (CRP) and 40 ng/mL for tumour necrosis factor-α (TNF-α). With a similar concept, another strategy was presented by Chen et al. wherein an affinity column (single straight microfluidic channel of 100 µm channel wide) was designed by imprisoning a specific antibody in a PA gel. As an example, follistatin (a 31.5 kDa glycoprotein) was introduced into the Ab-patterned PA-gel channel and finally separated electrophoretically according to the pore size, demonstrating a highly specific method compared to the method where the protein was only separated through size gel (Figure 2). This device allowed for the determination of both protein identification and size [69]. 

Compared with the conventional assays, their work demonstrated increased yields on recovery. Even better, the multistage assay gave power to both identification and quantification by fluorescence of the target protein without the use of surfactants. Despite these interesting results, there are still several improvements to be made, including multi-analysis in a parallel separation manner and enhanced automation. In addition, this type of approach might not be suitable for studying the conformation of proteins, as it has been shown that low electric fields could induce protein denaturation [70]. 

Another subdiscipline of microfluidics is droplet microfluidics, in which droplets serve as transport and reaction vessels. The principle of droplet formation is established on the use of two immiscible liquids (e.g., water-in-oil or oil-in-water) where the dispersed phase (water, the droplet) is injected into the flow of a continuous phase (oil, the medium in which droplets flow). The advantages of this technology are the size and quantity of picoliter droplets with faster and more precise reaction times compared to the level of single molecule in each droplet [71]. Furthermore, the droplets can provide an environment biocompatible for live cell experiments, ensuring advances in single studies [72,73] and the cultivation of cells [74]. For example, Utech et al. presented that alginate hydrogel droplets could provide a three-dimensional cell culture matrix, providing the nutrients and structural support required for the cells. After 15 days, an increased number of cells were observed in the microgel matrix [75]. Likewise, droplet-based screens can supply the platform for high-throughput experimentation with lower reagent use and cost and reduced operation steps [76,77]. 

Despite the microfluidics application genesis in the early 1950s, its clinical/industrial application has not been straightforward. If microfluidics has proved to be efficient for biomarker discoveries, the clinical and industrial applications remain confidential due to restrictive requirements of regulation organisms for clinical translation [60,78]. In particular, the material used for the fabrication of microfluidic chips play an important role in the microfluidic operability. Consequently, the choice of the material needs to be thoughtfully defined [23,63,79,80]. Nowadays, the materials used for the microfabrication of these microsystems can be divided into paper, inorganic, and polymer materials. The use of paper in the manufacture of microfluidics is interesting due to its low cost and absence of any pumps and valves since the liquid moves thanks to the capillary action. This could be very useful for bioassays in remote areas where there are limited resources. However, microfluidics based on paper has limitations. For example, the use of biological samples with surfactant is limited since the liquid can come out of the channels made in the paper and penetrate in the hydrophobic areas of the device, since outside of the channels, the paper is coated with hydrophobic material (e.g., wax), resulting in biological sample loss. Another limitation is its application for the analysis of a very low concentration due to the small LOD [81]. On the other hand, inorganic materials such as glass are widely used for the fabrication of analytical microchips essentially due to its good optical transparency, excellent chemical and mechanical stress resistance, and good surface properties. However, the manufacturing procedures are expensive and require specialized infrastructures since they are based on photolithography. Considering the price and easy fabrication, polymeric substrates have emerged as an alternative to glass in the fabrication of microchips used in bioanalytical applications [23,79]. Based on their physical properties, polymers can be subdivided into two major categories: elastomers and thermoplastics. Among the elastomer polymers, PDMS (Polydimethylsiloxane) is the most widely used for the fabrication of microfluidics devices, as it holds many attractive properties: (1) low cost, (2) ease of use, (3) good optical transparency, and (4) gas permeability. PDMS is an organic polymer that belongs to the silicones group, containing carbon and silicon in its structure. It is transparent in the visible range and is biocompatible for in vivo cell handling. The easy insertion of valves and pumps is possible due to the low modulus elasticity [63]. Usually, it is chosen to bind the PDMS to a glass substrate by oxygen plasma treatment (even if other different bonding methods exist) (Figure 3) [82]. This irreversible bonding enables the sealing of open channels in microfluidics [23,82,83]. 

Furthermore, in its natural state, PDMS is hydrophobic, and this can bring some problems, especially sample losses due to the unspecific adsorption of proteins onto the channel walls. After UV or oxygen plasma treatment, PDMS becomes more hydrophilic [63,84] making the surface of PDMS more resistant to the adsorption of hydrophobic and charged molecules, also avoiding the water and other solvents infiltrating into the PDMS substrate and consecutive PDMS deformation. For the thermoplastic subcategory, the most common materials are polystyrene (PS), poly(methyl-methacrylate (PMMA), polycarbonate (PC), and cyclic olefin copolymer (COC). In contrast to PDMS, these materials offer better solvent compatibility, but generally display a rigid structure and no permeation to gas, becoming a challenge to integrate for long term cell studies. In addition, the main costs and manufacture time are related to the production of moulds, which is not suitable for prototypes [23,60]. As a replacement for PDMS, some researchers have studied thiol-ene-based polymers. These polymers have a set of properties useful for microfluidics, namely a high stiffness that makes microfluidics suitable for high pressures, and a very limited absorption of biomolecules [85,86,87]. This is critical for on-a-chip integration with detectors and subsequent proteomic research. 

### 3.1. SEC, IP, and AC in Microfluidics

High demands in LC miniaturizations have been perceived due to the many advantages that microsystems provide, even if their implementations are still in their infancy. LC miniaturization using microfluidic devices provides gains for protein characterization, i.e., small injection volume, low peak dispersion, reduced flow rates, and enhanced sensitivity [80]. Other characteristics such the coupling of modules, integration of different functions in a single platform, chemical surface compatibility, ease of fabrication, and cost must be taken into consideration for the performance of an LC microfluidic chip. It is attractive for researchers to fuse different processes such as separation, extraction, pre-concentration, and desalting, among other actions, on a single platform for an automated analysis [67,88]. Different propositions have been published for the adequate incorporation of chromatographic resins such as SEC and functionalized beads and resins for IP and AC into microchannels (Figure 4). For example, SEC can be used after immunocapture beads as a strategy to ensure a highly pure and homogenous target protein, but the exact order of implementation can be varied to fit with experimental requirements.

The concept of a micro-fabricated SEC system has been explored and has grown in popularity, both for protein and cell separation. Nowadays, it can handle small amounts that are detected online, allowing for high-throughput analysis [44]. Millet et al. [88] proposed an enrichment of histidine (His)-tagged enhanced green fluorescent protein (eGFP) strategy through the connection of immobilized metal affinity chromatography (IMAC: affinity capture) and SEC for buffer exchange, facilitated by their novel 3D fluidic bridges design (Figure 5). 

This bridge design led to module integration, linked multiple moulds, and eliminated the necessity for new masks. To test the purity and concentration of eGFP, they used SDS-PAGE, and the authors found a separation quality approaching that of conventional IMAC column chromatography (275 μg of His-tagged eGFP from 773 μg crude *E. coli* cell lysate). Nonetheless, the high back pressures usually observed in packed particle microchannels can potentially result in common problems such as either leakage at the connectors or rupture of the columns on chips [80]. In a different configuration, Leong et al. produced a fritless microSEC channel with dimensions of 20 and 280 μm made by thiol-ene polymer, known as UV glue. Their approach allowed them to separate extracellular vesicles from proteins, due to the slower elution speed of the last ones. The use of this kind of glue allowed the use of a high pressure of up to six bars, an important parameter for a variety of detection methods [87].

Furthermore, the detection of low-abundance proteins present in complex biological fluids often requires an immunocapture step for target molecule enrichment [21]. The first attempts at immunocapture using microfluidics were engineered by affixing Ab on the channel walls [89]. Mohamadi et al. presented a microfluidic system where they imposed a hydrodynamic volume through a syringe pump at flow rates between 0.5 and 4 μL/min, resulting in an improvement in capture efficiency of synthetic βA (β-amyloid) peptides in CSF, the potential biomarker for the early diagnosis of Alzheimer’s disease. The immunocapture, re-concentration, and separation strategy allowed the capture of more than 90% of the added peptide in CSF and they were able to detect 25 ng of these synthetic βA peptides (βA-39, βA-40 and βA-41) in 100 μL of human CSF. Difficulties were encountered when they began to analyze real samples, in which no βA peptides were detected. To circumvent this problem, the immunocapture was accomplished off-chip and overnight in a 500 μL volume of CSF. Even if the Aβ1-40 subtype was detectable, the device was not able yet to detect all Aβ subtypes in CSF [35]. Seale and his team [52] isolated human serum albumin from a 4 μL droplet of serum sample by integrating a digital microfluidic with immunoprecipitation. This immunoprecipitation was carried out by a suspension of magnetic particles carrying Ab. After passing through the digital microfluidic module (based on an array of electrodes), the samples were then detected using the coupling of electrospray ionization and a triple quadrupole mass spectrometer operating in tandem (MS/MS) mode (Figure 6). Even if slight device modifications are required and more optimization needs to be performed, sample clean-up and targeted protein preconcentration were demonstrated to be promising for a wide range of applications in the future in biological, pharmaceutical, and medical product analysis.

### 3.2. Microfluidics for Biomarker Analysis

The identification of biomarkers in complex biological samples is challenging and requires advanced methods that are capable of separating without losing information about their structure and function. Breakthroughs in microfluidics have led to the search for disease-specific biomarkers for diagnostic and/or therapeutic applications. Due to its potential rapidity and real-time monitoring of samples, microfluidics serves as a strong tool to prevent and control the progression of infectious diseases. Different groups have presented examples of usage of microfluidic platforms for infectious disease detections. For example, Chen and his colleagues developed a PCR-based microfluidic approach in order to detect bacterial pathogens. The automated processing on the chip comprised a sequence of steps including sample introduction, cell lysis, nucleic acid isolation, and PCR amplification. The PCR product was applied to the sample pad of a lateral flow strip and captured. The strip was then scanned with the UPlink^TM^ reader [90]. In a different platform for POC testing, in order to simultaneously detect carcinoembryonic antigen and neuron-specific enolase biomarkers, a label-free microfluidic paper-based electrochemical aptasensor was developed. In clinical samples, this approach demonstrated high sensitivity and accuracy in the detection of both biomarkers due to the formation of the aptamer–antigen complex on the electrode surface [91]. Another group presented a microfluidic-based protein isoform assay for the quantification of the prostate-specific antigen isoform. Their strategy comprised the fabrication of a multistep single-channel immunoblotting to separate and immobilize protein isoforms and employ antibody probes. The results exhibited a five- to fifteen-fold gain in the time needed to perform the assay compared to the existing approach such as 2D electrophoresis with Western blotting [92]. Microfluidic technology devices also have provided an opportunity for pre-clinical drug development replacing animal models. In this context, the toxicity and metabolic response of drugs can be evaluated by the detection and analysis of specific biomarkers. Some good works are reviewed elsewhere [93].

### 3.3. Disease Potential Application: The Case of Parkinson Disease (PD)

PD diagnostics relies on the detection of clinical signs such as tremors, extreme tiredness, and memory loss. In addition, other diagnostics for PD rely on the detection of biochemical properties of alpha-synuclein (αS) and are linked to the increase in post-translational modifications, the modification of the ratio of soluble αS with respect to the oligomers, and the detection of truncated forms of alpha synuclein [94]. However, when these signals appear, the disease has attained a later stage with the loss of 50 to 60% of dopaminergic neurons [95]. For this reason, early detection of PD is of significant importance in order to have better therapy outcomes and open the possibility for new ones. Thus, early potential biomarkers for PD need to be found and characterized [96]. Detecting biomarkers from biological fluids (e.g., blood and urine) is a challenge due to (1) the complexity of the fluid samples, (2) low biomarker concentration and the fragility of cellular components that can result in structural changes, (3) defective uniformity of procedural conduct, (4) lack of statistical validation, and (5) miscommunication between organizations [60,78]. In this sense, strategies to address and surmount these points are needed. 

The structure–function relationship of a protein feeds a paradigm. On the one hand, for a determined function and in equilibrium, some proteins have a unique 3D structure. On the other hand, some proteins are highly dynamic in nature due to essential factors such as post-translational modifications that trigger structural modifications [60]. In fact, structural proteins can have heterogeneous ensembles of conformers in equilibrium such as alpha synuclein [96] or prion protein [97]. This conformational plasticity is of the utmost importance in several cellular processes: regulation of transcription/translation, and signalling pathway, etc. [98]. However, some conformational protein variations can be linked to physiological dysfunction [60,97]. For example, some evidence has proven that changes in the alpha-synuclein (αS) protein structure are associated with the pathogenesis of several neuro diseases, including PD. The predominant form of αS in equilibrium is the full-length protein. At some point, this equilibrium can be disrupted by different factors such as changes in the αS environment including pH, protein concentration, oxidative stress, chemical modifications, or mutations affecting its normal function. All of these factors are recognized to be associated with the pathogenesis [96]. This turning point is symbolized by the conversion from full-length state into an oligomeric state and ultimately forming aggregates that accumulate as fibrils, the principal component of Lewy bodies (see Figure 7). 

αS species, namely oligomeric assemblies, which have been considered the most neurotoxic species in the aggregation pathway, are thought to occur at the early stages of PD, causing the disruption of intracellular transport and consequently, degeneration of the neurons [96,99]. 

Consequently, due to this distinctive feature, αS can be considered as a potential biomarker to assess in a premature way whether or not the pathology exists and monitor its status [100]. Interestingly, αS present in the plasma [101] and red blood cells (RBC) [102] potentially can be used as biomarkers for the detection of PD as αS levels were found to be higher there than in the CSF (Figure 8). In this context, the use of these potential blood-based biomarkers can give rise to less-invasive sampling methodologies compared to the invasive technique needed for the collection of CSF, producing higher patient suffering.

For an inclusive structural-based diagnostic of the target, a need exists to develop biochemical techniques such as microfluidics that interfere in the least possible way with the target protein, while keeping specificity and sensitivity in the interest of obtaining reliable outcomes. The many advantages of microfluidics mentioned above as a means of separation and the improved sensitivity in detection of protein biomarkers (particularly when they are in extremely small concentrations) will play an important role in the detection of a disease in the early stages. 

In terms of microfluidics and especially when patients’ samples are used, this methodology would be of great help to limit sample handling and be compatible with very low amounts of material. The reliability, reproducibility, and validation of microfluidics methods is mandatory under these circumstances. Despite the difficulty of its commercialization and its clinical use, there are already successfully mature products in the market or in the late stage of development (reviewed elsewhere [105]). The introduction of microfluidic devices for clinical use is gradual but represents a reality [105,106]. The COVID-19 pandemic crisis is a good example of the fast approval of bead-based immune assays for diagnostics [107]. Micro- and nanotechnologies give unprecedented analytical performance to overcome some limitations of binding affinities to the attomole detection of biomolecules with optical detection [108]. Microfluidics is also of particular interest when it comes to analyzing biopsies that are very complex matrices [106]. From low sample volumes, scientists are able to sort, identify, and quantify circulating tumour cells [109]. The clinical implementation is gradually being carried out thanks to growing investments and circulating tumour cell miniaturized detections, which have already benefitted from official approval for diagnostics (CellSearch (Silicon Biosystems) [110] and Cobas EGFR Mutation test v2 (Roche Molecular System, Inc., Pleasanton, CA, USA)). 

## 4. Microfluidics Coupled to Mass Spectrometry

One of the first benefits that can be argued in favour of the microsystem scale is the notable decrease in sample volume as aforementioned. However, this also imposes some challenges in analyte detection, especially when the detectors are based on the interaction of the light with the analyte (as the size reduction results in optical path lengths reduction). In this regard, the use of highly sensitive detectors is demanded for a niche of microfluidics applications, since it is the detection method that dictates the sensitivity and selectivity of the method [80].

The most commonly used detectors in conjunction with microsystems have been optical/ultraviolet (UV), laser-induced fluorescence (LIF), and MS. Even though it is often depicted by its high cost and large dimensions of its equipment, MS has been offering numerous opportunities in medical application. It is highly sensitive, robust, and has the capacity to determine the authenticity, purity, and even structural details of unknown compounds. In addition, MS furnishes inherent miniaturization capabilities, being well adapted to microfluidic applications [80,88,111]. MS is globally constituted by an ionization source, an analyzer, and a detector. The emergence of native MS has allowed for the investigation of protein–protein and protein–ligand interactions, structure elucidations of biomolecular assemblies, determination of ligand or protein sub-unit stoichiometry, and has provided insights on conformational changes, topology, and dynamics of numerous proteins or protein complexes. This can be achieved since, upon appropriate conditions, the non-covalent interactions are preserved during the experiment [112]. In recent years, advancements in MS instrumental design have ensued, including the addition of soft ionization technology, the upgrading of computer systems conducted for a better know-how of analysis, increases in scan speed, reductions in detection limits down to ng mL^−1^ or pg mL^−1^, and the adoption of low-flow rates [80,113] as compared to the first MS devised by Aston in 1919 [114]. The most widely used technique to produce ions used in native MS is electrospray ionization (ESI) [115]. The ionization is soft in the sense that little or no fragmentation occurs (just a little residual energy is retained by the analyte) and non-covalent interactions such as electrostatic and hydrogens bonds can be preserved in the gas phase. ESI leads to the production of multiple charge states from liquid samples at atmospheric pressure, allowing the analysis of large proteins or complexes, under experimental conditions close to physiological ones that preserve protein structure when transferred in the gas phase. The desorption–ionization of samples is performed by an ESI source where the sample is subjected to the application of a high voltage (of a magnitude of 1 to 4 kV) at the end of a metal ESI capillary [116]. A cone shape, known as the Taylor cone, begins to form at the capillary tip where charge droplets are emitted. These droplets will then undergo some evaporation and Coulombic explosions that will lead, in the final stage, to the production of a multiple-charged ion in the gas phase (Figure 9b). It is important to note here that the gas phase structure of a protein does not necessarily reflect the structure in solution. However, some studies report that the gas phase structure retains solution structure characteristics [97,117].

ESI when coupled with a high accurate analyzer has the capacity to obtain stoichiometric information. The miniaturization of ESI systems has shown increases in sample ionization efficiency in MS and thus, improvements in sensitivity. Furthermore, the use of these miniaturized ESIs has allowed for working at low-flow rates (i.e., 10–100 nL/min), entitled as nanoESI [116], making it possible to couple to systems that work with rates of the same order of magnitude, such as microfluidics [83,118]. The coupling of this analytical method with microfluidics devices (Figure 9a), given the low sample volume handled, completely changed how the -omic fields can be explored as the coupling, benefits from both technologies. In practice, however this interface presents a significant challenge due to high back pressure that can cause damages to the microfluidic systems as well as leakage at the connectors [80], especially in SEC-based microfluidics where the channel pressure is significant. Recently, this coupling has been essentially based on the use of nanospray-fused silica capillary emitters, the earliest miniaturization of emitters that were inserted into the microfluidic system [83], enabling the decrease in dead volumes and thus, the enhancement of the low-level species detection in complex samples. To overcome this problem, scientists presently favour sheathless interfaces. The high potential (kV) required for the ESI process also requires electrical integration of the microfluidic system. Currently, coating the spray emitter with conductive materials, namely gold or conductive polymers as well as implanting a wire at the emitter tip, has been the chosen strategy to achieve successful electro-nebulization. However, the implementation of these strategies brings with them several problems, such as low mechanical robustness, poor reproducibility, and unstable electrospray. To solve this issues, sheathless interfaces using a chemically etched porous emitter have been applied, resulting in a much-improved performance [119].

Employing an LC-MS/MS approach, the separation of tryptic protein digests of bovine serum albumin was accomplished by Yin’s group. The separation was performed using conventional reversed-phase columns (a class of adsorption-based chromatography). The approach resulted in sub-femtomole sensitivity, minimal carryover of the sample, and robust and stable electrospray, achieved by the low-flow rates between 100 and 400 nL/min and by elimination of fluidic connections (commonly applied between each fluidic component and nano-electrospray tip) [120]. In 2011, Chambers et al. [121] projected a glass microfabricated platform based on two-dimensional separation using reversed-phase LC-CE modules with integrated ESI for MS, i.e., LC-CE-ESI-MS system. This microfluidic device allowed the use of low-flow rates without being subject to substantial band broadening, an important point that needs to be considered. Tryptic peptides from bovine serum albumin (BSA) and *E. coli* cell lysate were successfully analyzed and exhibited the potential of this approach. Nevertheless, reversed-phase LC might not be suitable for structural analysis due to the necessity of a high concentration of organic modifiers that denature the proteins. Another group [30] demonstrated improvements in terms of sensitivity using a tryptic digest of haemoglobin. The detection limit obtained was less than five red blood cells. From their work, a multi-nozzle emitter array chip for on-a-chip and online LC-nanoESI-MS analysis was presented in favour of small-volume proteomics (Figure 10). The electric field at the nozzle tip was improved by sharpening the emitter system with an angle of ~20°. Further enhancements should be attained by increasing their nozzle numbers and by essentially optimizing their geometry and sharpening parameters. In addition to sensitivity improvements (via multi-nozzle emitters), high-throughput was obtained during their experiments (via multi-channel). These technical upgrades presented by the different research groups considerably lowered the sample consumption and enhanced the limit of detection. 

Integrating different chromatographic components on a single microsystem and interfacing them with ESI-MS by an electrospray emitter has been a major trend. This type of configuration for carrying out complex multistep assays enables an easy sample preparation and an automated protocol without the commitment of manual intervention (note that the microchannel design must be developed to achieve the desired features, e.g., the use of micromechanical valves). In addition, there is an increased protein separation capability, providing a crucial system for high-throughput proteomic studies. Nevertheless, this is a strategy that is still under intensive development and thus, there are several pitfalls and limitations that the user needs to be aware of. First, the purity of the protein must be high enough to prevent spectrum interpretation difficulties. Thus, analyte preconcentration needs to be improved. Multidimensional separations in the same workflow are desired in order to decrease the complexity of the samples and purification of the target analyte. SEC, AC, and IP are good base methods thanks to their ability to separate the analyte by size (SEC) and specificity of Ab-Ag interaction (AC and IP). Second, the high back pressure needed to inject the solutions to MS detectors can cause damage to microchannels. Thus, preference is given to microfluidics made of materials strong enough to withstand high pressures [80]. 

Another problem arises when dealing with conformers in equilibrium: an ion can be represented by several conformers if they have an identical mass, but different structures. The merging of ion mobility (IM) with MS supplies an opportunity to distinguish conformer ions based on an additional dimension known as collision cross sections (CCS) [122,123]. In this technique, the conformers are separated according to the frictional force that is proportional to the accessible surface of the ions to the oppositely moving drift gas. More precisely, the gaseous ions derived from the sample are separated based on their charge, shape, and size [123]. Therefore, larger ions (with larger CCS values) will move slower as a result of greater collisions with the buffer gas (Figure 11c, orange) compared to smaller ions (Figure 11c, green).

The coupling of IM-MS with separation methods has opened the possibility to understand the structural changes of different analytes critical to the cellular fate. Mironov and co-workers explored the coupling of IM-MS with capillary electrophoresis and UV detection, observing the structural changes in human tissue transglutaminase. This study allowed a real-time detection of protein conformational isomers changes and activity measurement by following the enzymatic product formation in a single experiment. They showed that for the same protein under study, there were two conformers due to differences in collision cross-section (CCS) values [27]. Wyttenbach and Bowers also relied on ionic mobility technology to study whether there would be structural differences in ubiquitin protein after the solution has passed through ESI and IM-MS analysis. They concluded that the native structure of the protein is preserved but it was necessary to adjust several parameters, mostly in the MS, such as the collision energy of ions or time scales, in order to avoid Coulomb repulsions and therefore unfolding [117,124,125]. 

Understanding this conformation modulation can open new doors for therapeutic applications of proteins presenting structural dynamics, since the implementation in protein research of microfluidics coupled to ESI-IM-MS contributed to significant advancements, namely faster and cheaper analysis. Such technology will be essential for the support of proteomic and de novo biomarker discovery [126]. However, only a limited number of studies have been published to date on the application of microfluidics ESI-IM-MS for the investigation of biomarkers. For isomers that have quite tricky structural differences, this technique has some limitations so far concerning complete isomer separation. Other different strategies have been proposed and developed such as the insertion of deuterium reagents into the mobility cell in order to form a microdroplet hydrogen–deuterium exchange that gives better intricate structural details [127]. There is still substantial optimization to carry out, although there is a clear trend toward the development of multifunctional chip-ESI-IM-MS interfaces, not just focusing on separation strategies. 

## 5. Conclusions and Future Perspectives

Early diagnosis grants early-adapted treatments, furnishing better chances to improve a patient’s health as compared to ill-informed procedures realized while waiting for clinical results. Now that low-abundance biomarkers and isoform bioanalysis have led to an increase in several studies, the fine tuning of protein purification methods without affecting the protein’s structure becomes increasingly more important. Microfluidics has been a topic of interest to the scientific community for several decades. Its capacity to measure biomarkers in small quantities and even single cells samples that are not measurable using conventional technologies, makes it notably appropriate for peptidomic studies. However, its widespread implementation is yet to be seen and has made little progress towards becoming a real-world product. Possible reasons include the procedures to fabricate chip-based LC in the laboratory, the coupling with other interfaces such detectors, or sample pre-treatment steps. A great deal of effort needs to be devoted to the purification of proteins for structural studies and testing of their biological properties. Conformation is often forgotten since primary structures and concentration are easier to obtain. However, the conformational states are of critical importance to discover new biochemical mechanisms involved in major diseases.

In the forthcoming years, it is predictable that further work will continue in the creation of improved microfluidics, especially for biomarker purification. The combined use of these miniaturized AC, IP, and SEC modules with ESI-IM-MS is also expected, notably in -omic fields. This type of approach has presented potential applications in the discovery of new compounds and biomarkers. To date, the majority of literature has been focused on the coupling of microfluidic chips to ESI-MS. There are fewer articles on the development of microfluidic chips for interfacing with ESI-IM-MS. The practical employment of these platforms for clinical diagnosis is still challenging and the addition of IM can provide a novel interesting strategy.

Furthermore, advancements are expected for the nanoESI-MS technology to afford better sensitivity and analysis. Important investments that bring together scientists, engineers, and business experts are needed for microfluidics to reach its full potential.

## Figures and Tables

**Figure 1 proteomes-11-00019-f001:**
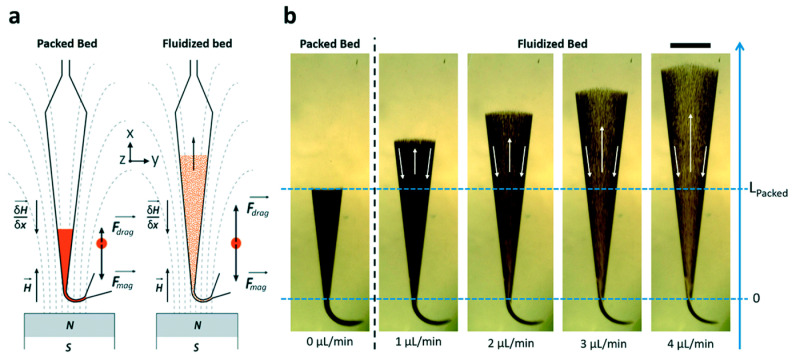
Behaviour of magnetic beads fluidization subjected to an external applied field and flow rate. (**a**) Schematic representation exhibiting the orientation and the forces of magnet streamline. (**b**) Pictures exhibiting the packed and fluidized bed for different flow rates. Both position of magnet and flow rate play a major role in the immunocapture of the analyte target (scale bar = 1 mm). Reprinted with permission from [65]. Copyright 2023 The Royal Society of Chemistry 2017.

**Figure 2 proteomes-11-00019-f002:**
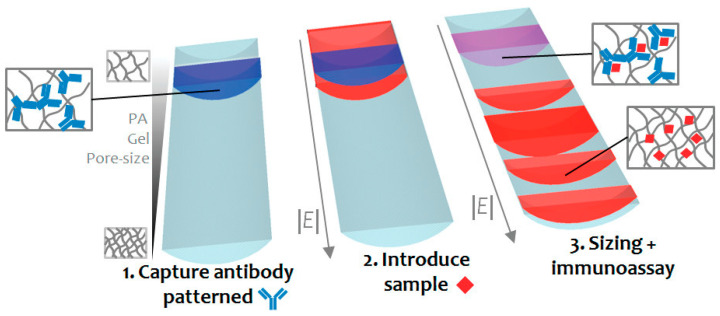
Protein size and immunoaffinity separation in a straight microfluidic chip. Reprinted with permission from [69]. Copyright 2011 American Chemical Society.

**Figure 3 proteomes-11-00019-f003:**
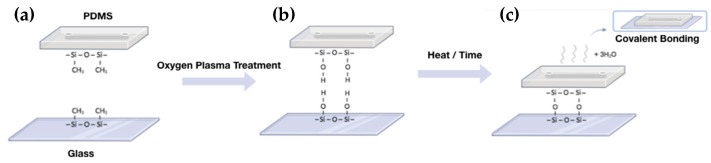
Overview of PDMS glass-bonding process. (**a**) Both substrates, PDMS and glass, are coated by repeating units of –O–Si– (CH3)2–. (**b**) After surface activation by oxygen plasma treatment, the hydrocarbon group are eliminated resulting in silanol groups (–Si–OH). (**c**) Under a certain time or temperature, water molecules are lost, enabling a strong covalent siloxane bond (Si–O–Si) between the two substrates. For further information, see reference [82].

**Figure 4 proteomes-11-00019-f004:**
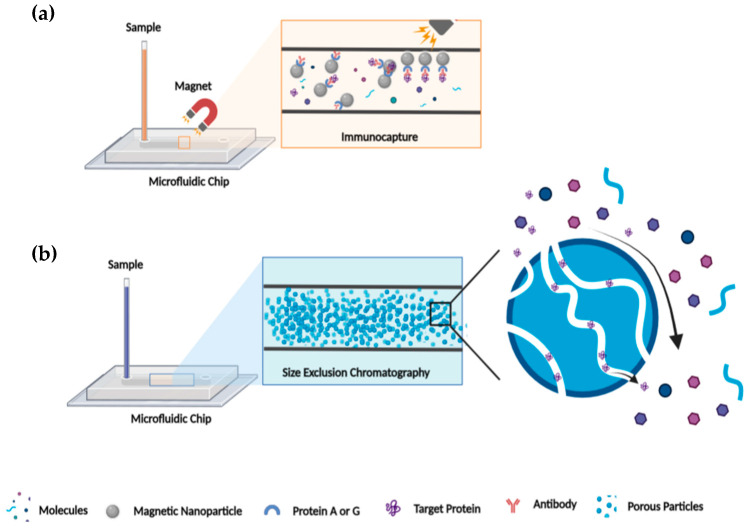
Schematic illustration of IP and SEC microfluidic chip. (**a**) IP illustration in microfluidic chip: protein A/G covalently linked on magnetic bead in the immunocapture channel of microfluidic chip allowed capturing the antigen/antibody complex and consequently the target protein. (**b**) SEC illustration in microfluidic chip: target protein separation from “salt” by size exclusion through porous particles that fills the microfluidic channel.

**Figure 5 proteomes-11-00019-f005:**
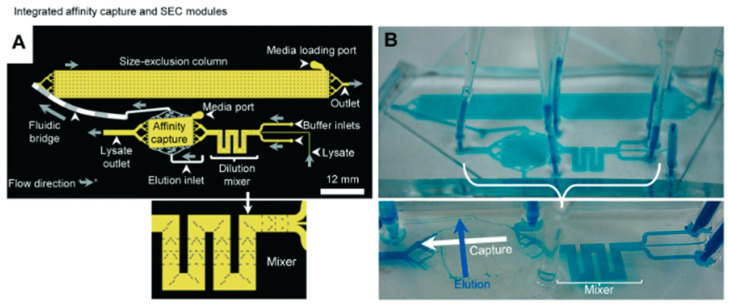
Purification of His-tagged eGFP by IMAC followed by desalting through SEC. (**A**) Schematic of the IMAC and SEC modules (2D layout) connected by fluidic bridges. Sample flow direction is also shown (grey arrows). (**B**) A photo of microfluidic chip with blue dye continuously flowing into the channels. Reprinted and adapted with permission from [80]. Copyright 2015 The Royal Society of Chemistry.

**Figure 6 proteomes-11-00019-f006:**
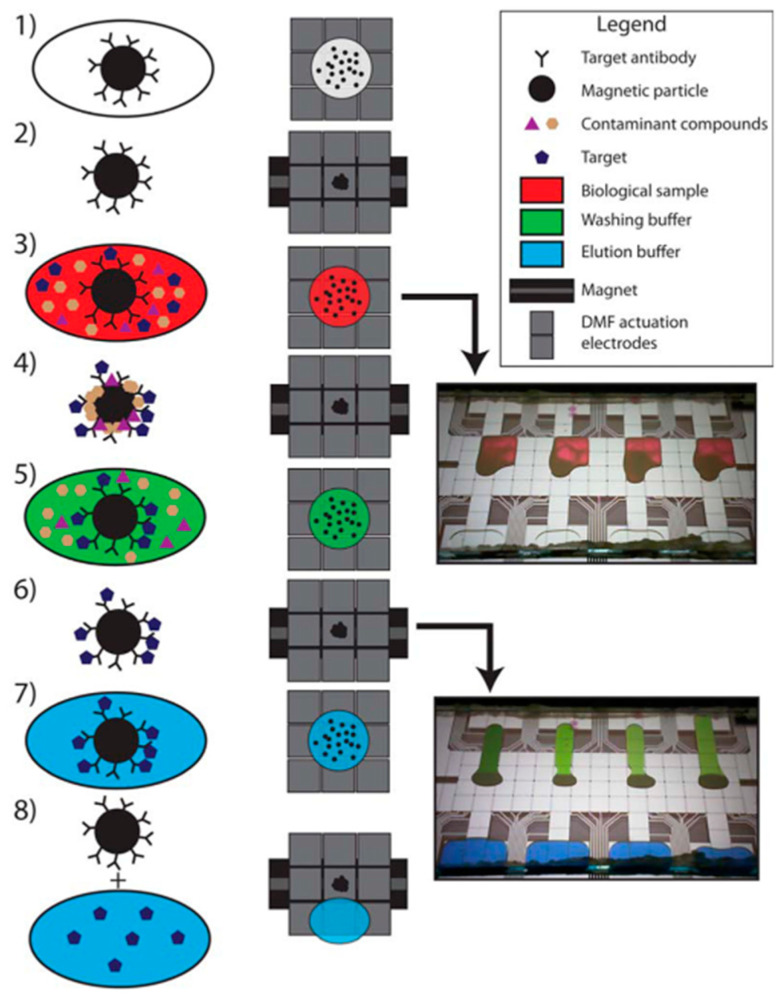
Steps in the immunopurification process by digital microfluidic chip. (1) Magnetic NPs functionalised with Ab are introduced in the chip where microfluidic electrodes allowed the droplet shifting in the different compartments. (2) The magnetic beads are concentrated using magnets. (3) The biological sample is introduced and (4) the complex is formed between the target molecule (albumin protein) and the functionalized NPs. (5) Washing steps are performed to eliminate contaminants and (6) the complex is enriched by the magnet. (7) To break this complex, the elution buffer is added and (8) the elute is separated from the NPs using again a magnet and recovered for posterior MS analysis. Reprinted with permission from [52]. Copyright 2016 American Chemical Society.

**Figure 7 proteomes-11-00019-f007:**
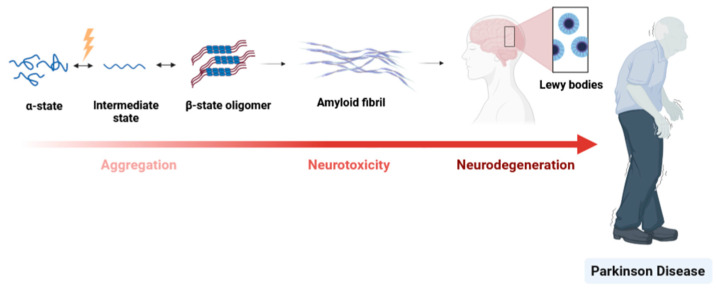
PD dementia. The key event in PD is misfolding of the unfolded αS protein monomer into an isoform that is rich in α-sheet structure. This conformational change may result in the formation of Lewy bodies. This abnormal aggregation of proteins—that develop inside nerve cells—is a hallmark of PD.

**Figure 8 proteomes-11-00019-f008:**
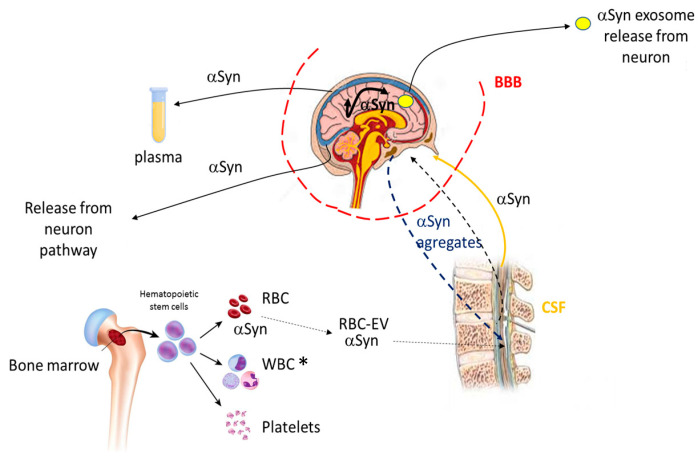
αS trafficking and potential PD biomarkers. αS can be found in different cells and body fluids, communicating between them. The αS either in its soluble form or in aggregates is transported bidirectionally, from the brain to the CSF and vice versa through the blood–brain barrier (BBB). αS can also join the CSF through the formation of extracellular vesicles (EVs) from red blood cells (RBC). αS can also move from one compartment to another inside the brain. Finally, the protein can be released from the brain following the neuron pathway or exosome release from neurons. The αS is also found in plasma. For further information, see references [101,103,104]. * WBC (white blood cells).

**Figure 9 proteomes-11-00019-f009:**
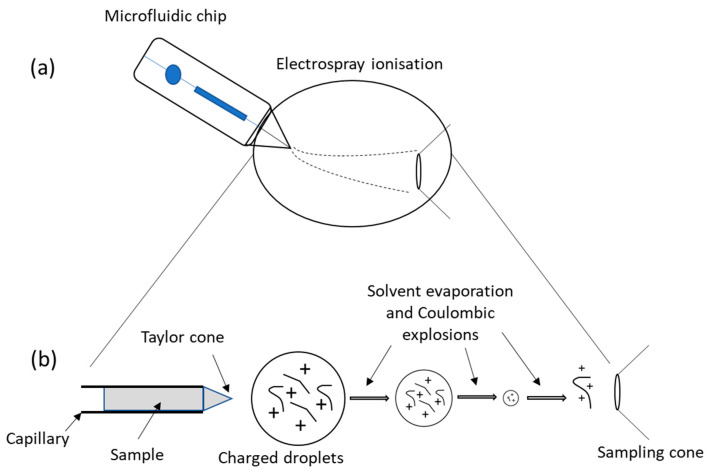
Schematic representation of the microfluidic chip-ESI-MS coupling. (**a**) Microfluidics containing the sample is placed adjacent to the inlet side of ESI with the formation of a spray that is then driven toward the MS. (**b**) The formation of the Taylor cone from which a jet of charged particles emerges above a threshold voltage. For further information, see reference [83].

**Figure 10 proteomes-11-00019-f010:**
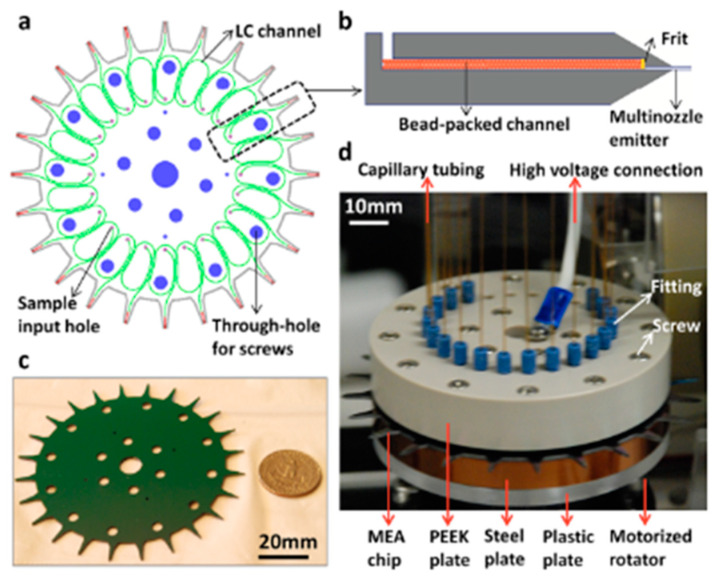
Overview of chip LC-MS system for small volume samples. (**a**) The sample is introduced through the input holes (small holes in purple) that is then separated in the LC bead-packed channel (in green). For the identification and quantification of proteins and peptides, the separated sample is injected in nanoESI-MS via multi-nozzle emitter (in red). (**b**) Illustration of LC channel showing the beads packed and retained with a frit. (**c**) Size comparison between the chip and the United States coin worth 25 cents (scale: 20 mm). (**d**) Photograph showing the chip-MS coupling by capillary tubing (scale: 10 mm). Reprinted with permission from [30]. Copyright 2013 American Chemical Society.

**Figure 11 proteomes-11-00019-f011:**
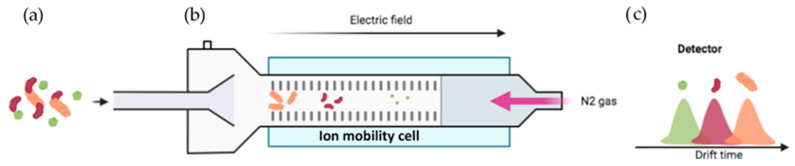
Working principle of IM-MS. (**a**) The complex samples are constituted by conformers that have the same mass with slight variations in their structure. The sample is injected in MS and ionized by ESI. (**b**) Under an electric field and an opposite buffer gas flow, such as N_2_, the ions are separated based on their shape, charge, and size. (**c**) The more compact conformers (in green) are first detected, and the open ones (in orange) are detected last. For further information, see reference [123].

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
