# Peer review of "Proteomics Methodologies: The Search of Protein Biomarkers Using Microfluidic Systems Coupled to Mass Spectrometry"

_proteomes, 2023, doi:10.3390/proteomes11020019_

Round 1

Reviewer 1 Report

- the title should be rephrased

- the references should be updated with original proteomics previous works dealing with LC-chip MS and IMS.

- lines 53-56; SEC conditions might have an impact on protein aggregation.

- lines 73-75: miniaturization is not characterized by high resolution 

- sections 2 and 3 are long and not related with the topic of microfluidics

- in section 4, droplets-based microfluidics should be considered

- section 4.2 should be deleted since there is no microfluidics applications

- in section 4, the authors should discuss about the reliability/reproductibility/validation of microfluidics methods. Are those methods really compatible with the analyses of patient samples that are most of the time difficult to obtain and cannot be wasted?

- The conclusion should be further developed with the authors' opinion on the next necessary developments in microfluidic devices and microfluidics methodology in proteomics.

Author Response

We thank reviewer 1 for his remarks and ways suggested to improve our manuscript. We followed the recommendations of reviewer 1 and we hope that these last will fit the requirements.

- the title should be rephrased

We rephrased the title that seems to fit better the content of the article. We hope that this last is now in agreement with reviewer 1 recommendation.

Title : “Proteomics methodologies: the search of protein biomarkers using microfluidic systems coupled to mass spectrometry.”

- the references should be updated with original proteomics previous works dealing with LC-chip MS and IMS.

The update of the bibliography with references of original proteomics work with LC-chip MS and IMS was made.

- lines 53-56; SEC conditions might have an impact on protein aggregation.

SEC is the standard method for protein aggregate analysis. The technique involves passing molecules through a column containing porous polymer or silica beads. The choice of pore size is related to the size of the molecule to be separated. For the separation of mAbs and their aggregates, this is around 300Å.  Molecules are separated based on their hydrodynamic volume. Given the sensitivity and reproducibility of the method, SEC may be considered as the standard method for monitoring protein aggregation [114] and is included in the list of the typically used tests in the European Pharmacopoeia guidance document entitled “Technical Guide for the Elaboration of Monographs on Synthetic Peptides and Recombinant DNA Proteins,” 1st Edition 2006.[115]. See: ttps://www.ncbi.nlm.nih.gov/pmc/articles/PMC3556795/.

Since SEC is used to detect aggregates in the European Pharmacopoeia guidance we can say that if SEC would promote the formation of aggregates this technique will be give up. In addition, we perform SEC analyses for more than 8 years now and we never observed any artifactual aggregates when using this technique. Would you please find enclosed two of our articles supporting our statement.

Size Exclusion Chromatography-Ion Mobility-Mass Spectrometry Coupling: a Step Toward Structural Biology. Van der Rest G, Halgand F. J Am Soc Mass Spectrom. 2017 Nov;28(11):2519-2522. doi: 10.1007/s13361-017-1810-0. Epub 2017 Sep 20.

Transient multimers modulate conformer abundances of prion protein monomer through conformational selection. Van der Rest G, Rezaei H, Halgand F. Sci Rep. 2019 Aug 21;9(1):12159. doi: 10.1038/s41598-019-48377-w.

- lines 73-75: miniaturization is not characterized by high resolution.

We fully agree with reviewer 1 and changes were made accordingly in the text.

- sections 2 and 3 are long and not related with the topic of microfluidics

These sections were written to introduce the chromatographic methodologies and their limitations before discussing the pros and the cons of miniaturization. We believe that these two sections are important to understand the interest of using a lab on chip.

- in section 4, droplets-based microfluidics should be considered

We made bibliography and extend this section.

- section 4.2 should be deleted since there is no microfluidics applications

We disagree with reviewer 1, since the choice of Parkinson disease for the search of biomarkers is a good example of the interest of developing a chip coupled to ion mobility and mass spectrometry, especially for the detection of protein conformers. However, we added another paragraph before the Parkinson section to show examples of the interest of Lab on chip for proteomics and biomarker discovery.

- in section 4, the authors should discuss about the reliability/reproductibility/validation of microfluidics methods. Are those methods really compatible with the analyses of patient samples that are most of the time difficult to obtain and cannot be wasted?

We fully agree with reviewer 1. Bibliography was added as well as comments on reliability, reproductibility and validation of microfluidics methods.

- The conclusion should be further developed with the authors' opinion on the next necessary developments in microfluidic devices and microfluidics methodology in proteomics.

The conclusion was improved in this way.

Reviewer 2 Report

The review paper is a very well-written article about the current state of microfluidics in the disease biomarker discovery field, and the highlights of advantages and disadvantages of the use and power of mass spectrometry is clearly written. The example of disease a biomarker for Parkinson's disease and the discussion of protein isoforms and conformers are extremely important clinccally, and this manuscript highlights the topics very well. There are suggestions for minor grammatical changes, submitted as a PDF file with markups and comments.

Author Response

We would like to warmly thank reviewer 2 for the reviewing of our article. Requested corrections were made in the text.

Reviewer 3 Report

This paper provides a comprehensive overview of different protein enrichment techniques using miniaturized devices and their coupling to mass spectrometry for the identification of protein biomarkers in diseases. The authors presented a well-structured and logically sequenced paper with clear and concise writing that is easy to follow. The paper begins by discussing various proteomics methodologies currently available, including different fractionation techniques that preserve the native structure of proteins, such as X-ray diffraction, nuclear magnetic resonance, and Foster resonance energy transfer, as well as gel-based and mass spectrometry-based chromatography. The authors highlight the strengths and weaknesses of each method in protein separation, and their arguments are supported by relevant references. Furthermore, the authors discuss the advantages of size-exclusion chromatography and affinity chromatography in protein separation, their experimental conditions, and the limitations associated with them in the third section of the paper. They also highlight the recent trend of microfluidics devices as a potential solution to the challenges of sample pre-treatment and analysis. The authors describe the advantages and challenges of different microfluidic systems and summarize improvements in microfluidics as a potential biochemical technique for the detection of protein biomarkers in Parkinson's Disease.  Mass spectrometry (MS) is the standard analytical tool for detecting biomarkers due to its high sensitivity, but the biomarkers must be pure enough to avoid chemical noise. Therefore, the coupling of microfluidics to MS has been increasing for biomarker discovery, and the paper discusses the different approaches for protein enrichment using microfluidics coupling to MS, and their advantages, and disadvantages of each method. I believe that this paper has the potential to make a valuable contribution to the field of microfluidics.

Minor corrections:

Overall sections

The information is presented logically, and the transitions between different topics are smooth. The language used is appropriate, and technical terms are defined appropriately. However, some sentences are too long and can be confusing. It may be useful to break them down into smaller sentences to improve readability. For example, in section 3, the phrase in sentence 202-206, and lines 214-218, could break down into smaller sentences.   

Section 4

The authors provide detailed descriptions of several methods to improve microfluidic chips, however, the authors could provide more specific examples of biomarkers and drugs that have been discovered using microfluidics to provide more concrete evidence for the potential of this technology.

Section 5

In line 666, “This type of configuration enables an easy sample preparation and an automated protocol without the commitment of manual intervention.” Please elaborate on how easy the sample preparation is compared to other techniques. 

Figures:

Overall, the figures are well presented and support the arguments throughout the text. However, some figures need a figure legend (for example figure 6), and the texts in Figures 7 & 8 should be enlarged for easy reading (inconsistent, some texts are larger than others). According to the text and figure title in Figure 9, it should be labeled as Figure 9 A and 9B.   

Author Response

Many thanks to reviewer 3 for extended reviewing of our article.

Minor corrections:

Overall sections

The information is presented logically, and the transitions between different topics are smooth. The language used is appropriate, and technical terms are defined appropriately. However, some sentences are too long and can be confusing. It may be useful to break them down into smaller sentences to improve readability. For example, in section 3, the phrase in sentence 202-206, and lines 214-218, could break down into smaller sentences.   

The text has been checked to remove the sentences that are too long and rephrase these last, as well as an extensive English correction by a native English-speaker.

Section 4

The authors provide detailed descriptions of several methods to improve microfluidic chips, however, the authors could provide more specific examples of biomarkers and drugs that have been discovered using microfluidics to provide more concrete evidence for the potential of this technology.

According to the suggestion of reviewer 3, we added a section showing examples of biomarkers and drugs discovery in the field of microfluidics.

Section 5

In line 666, “This type of configuration enables an easy sample preparation and an automated protocol without the commitment of manual intervention.” Please elaborate on how easy the sample preparation is compared to other techniques. 

This part has been improved to show the advantages of automated sample preparation.

Figures:

Overall, the figures are well presented and support the arguments throughout the text. However, some figures need a figure legend (for example figure 6), and the texts in Figures 7 & 8 should be enlarged for easy reading (inconsistent, some texts are larger than others). According to the text and figure title in Figure 9, it should be labeled as Figure 9 A and 9B.   

Corrections were made according to reviewer’s 3 recommendations.

Round 2

Reviewer 1 Report

The modifications improved significantly the manuscript.